# Does the Intensity of Therapy Correspond to the Severity of Acute Respiratory Distress Syndrome (ARDS)?

**DOI:** 10.3390/jcm13237084

**Published:** 2024-11-23

**Authors:** Domenico Nocera, Stefano Giovanazzi, Tommaso Pozzi, Valentina Ghidoni, Beatrice Donati, Giulia Catozzi, Rosanna D’Albo, Martina Caronna, Ilaria Grava, Gaetano Gazzè, Francesca Collino, Silvia Coppola, Simone Gattarello, Mattia Busana, Federica Romitti, Onnen Moerer, Michael Quintel, Luigi Camporota, Luciano Gattinoni

**Affiliations:** 1Department of Anesthesiology, University Medical Center Göttingen, Robert-Koch-Str. 40, 37075 Göttingen, Germany; domenico.nocera@studio.unibo.it (D.N.); s.giovanazzi001@unibs.it (S.G.); valentina.ghidoni.27@gmail.com (V.G.); donati.beatrice92@gmail.com (B.D.); giulia.catozzi@unimi.it (G.C.); caronnamartina@gmail.com (M.C.); ilaria.grava@unito.it (I.G.); gaetano.gazze@gmail.com (G.G.); mat.busana@gmail.com (M.B.); fromitti@icloud.com (F.R.); onnen.moerer@med.uni-goettingen.de (O.M.); mquintel@gwdg.de (M.Q.); 2Department of Medical and Surgical Sciences, Alma Mater Studiorum, University of Bologna, Via Massarenti 9, 40138 Bologna, Italy; rosannad_7@hotmail.it; 3Department of Medical and Surgical Specialties, Radiological Sciences and Public Health, University of Brescia, Piazzale Spedali Civili 1, 25121 Brescia, Italy; 4Department of Health Sciences, University of Milan, Via Festa del Perdono 7, 20122 Milano, Italy; tommaso.pozzi94@gmail.com; 5Department of Anesthesia and Intensive Care, ASST Santi Paolo e Carlo, San Paolo University Hospital, 20142 Milan, Italy; silvia_coppola@libero.it; 6Department of Health Science, Department of Anesthesia and Intensive Care, AOU Careggi, Largo Brambilla 3, 50139 Firenze, Italy; 7Department of Anaesthesiology, Critical Care and Pain Medicine, “Sapienza” University of Rome, 00199 Rome, Italy; 8Department of Anesthesia, Intensive Care and Emergency, AOU Città della Salute e della Scienza di Torino, Corso Bramante 88, 10126 Torino, Italy; francesca.collino@unito.it; 9Department of Anesthesia and Intensive Care Medicine, IRCCS San Raffaele Scientific Institute, 20132 Milan, Italy; gattarello@gmail.com; 10Centre for Human & Applied Physiological Sciences, School of Basic & Medical Biosciences, King’s College London, London WC2R 2LS, UK; luigi.camporota@kcl.ac.uk; 11Guy’s & St Thomas’ NHS Foundation Trust, London SE1 7EH, UK

**Keywords:** ARDS severity, mechanical power, respiratory mechanics, ventilator-induced lung injury, gas exchange, ARDS

## Abstract

**Objectives:** The intensity of respiratory treatment in acute respiratory distress syndrome (ARDS) is traditionally adjusted based on oxygenation severity, as defined by the mild, moderate, and severe Berlin classifications. However, ventilator-induced lung injury (VILI) is primarily determined by ventilator settings, namely tidal volume, respiratory rate, and positive end-expiratory pressure (PEEP). All these variables, along with respiratory elastance, are included in the concept of mechanical power. The aim of this study is to investigate whether applied mechanical power is proportional to oxygenation severity. **Methods:** We analyzed 291 ARDS patients (71 mild, 155 moderate, and 65 severe). We defined low, middle, and high mechanical power by dividing the entire population into tertiles with a similar number of patients. In each oxygenation class, we measured computed tomography (CT) anatomy, gas exchange, respiratory mechanics, mechanical power, and mortality rate. **Results:** ARDS severity was proportional to lung anatomy impairment, as defined by quantitative CT scans (i.e., lung volume and well-aerated tissue decreased across the ARDS classes, while respiratory elastance increased, as did mortality). Mechanical power, however, was similarly distributed across the severity classes, as the decrease in tidal volume in severe ARDS was offset by an increase in respiratory rate. Within each ARDS class, mortality increased from low to high mechanical power (roughly 1% for each J/min increase). **Conclusions:** Both lung severity and mechanical power independently impact mortality rates. It is tempting to speculate that ARDS severity primarily reflects the natural course of the disease, while mechanical power primarily reflects the risk of VILI.

## 1. Introduction

Traditionally, respiratory treatment in ARDS (acute respiratory distress syndrome) has focused primarily on achieving and maintaining adequate oxygenation. Despite over 50 years [1] of evolving strategies, the intensity of respiratory support is still largely guided by the degree of oxygenation impairment. Recent ARDS guidelines from the ESICM (European Society of Intensive Care Medicine) [2] and the ATS (American Thoracic Society) [3] continue to recommend interventions based on the severity of oxygenation impairment (mild, moderate, or severe) with indications of strategies, such as high PEEP (positive end-expiratory pressure), prone positioning [4], and extracorporeal support [5,6], all based on these oxygenation thresholds.

However, there is growing recognition that the potential harms of mechanical ventilation are more closely related to the management of CO_2_ (carbon dioxide) clearance and respiratory mechanics, rather than on the oxygenation levels alone. Strategies like permissive hypercapnia [7] and lung-protective ventilation [8] aim to minimize lung damage caused by mechanical factors. The risk of ventilator-induced lung injury (VILI) is largely influenced by the interaction between underlying lung pathology [9,10], and the consequent reduced “baby lung” size, the high elastance, and the ventilator settings chosen by the clinician [11].

This interaction can be summarized in the concept of “mechanical power” [12], which integrates the primary factors predisposing the lungs to injury (e.g., high elastance) with the ventilator settings (e.g., tidal volume, respiratory rate, and PEEP). If the degree of oxygenation impairment was directly proportional to the risk of VILI, as measured by mechanical power, tailoring the intensity of therapy based on oxygenation levels or mechanical power would yield similar outcomes. However, our recent study involving patients with moderate to severe ARDS found that the intensity of mechanical power was similar across different oxygenation severity categories. This led us to conclude that the risk of VILI is more closely linked to mechanical variables than to oxygenation levels.

In the present study, we extended our analysis to include patients with mild ARDS, focusing on mortality rates. The primary hypothesis is that oxygenation impairment reflects the natural progression of ARDS and its associated mortality risk, while mechanical power, primarily dependent on ventilator settings, has a stronger influence on mortality across all severities of ARDS (mild, moderate, and severe). If this hypothesis holds true, it suggests that the choice of respiratory therapy should be based less on the severity of oxygenation impairment and more on the mechanics of the respiratory system.

## 2. Methods

### 2.1. Study Population

The study population includes 291 patients with ARDS, enrolled between May 2003 and July 2024, from San Paolo University Hospital of Milan (Italy), University Medical Center Göttingen (Germany), Ospedale Maggiore Policlinico of Milan (Italy), Azienda Ospedaliera San Gerardo Monza (Italy), and Pontificia Universidad Catolica de Chile (Chile). Patients were classified based on their severity.

Quantitative CT scan taken in standard conditions and respiratory and gas exchange variables collected within 24 h from entry were prerequisites for enrollment and study inclusion. Patient with respiratory failure due to COVID-19 were excluded from this analysis due to their prevalent vascular pathology in the early stage of the disease.

This study is a retrospective analysis of the combined Milan and Göttingen databases. The ethics committee was notified and permission to use the data was granted (Göttingen Antragsnummer 14/12/12).

### 2.2. Study Design

Population Stratification

The patient population was stratified into the following groups:**ARDS Severity_PF_ classes**

Using baseline records, we classified the patient population according to the Berlin definition into the following three severity groups: mild ARDS (P/F ratio between 201 and 300 mmHg, n = 71), moderate ARDS (P/F ratio between 101 and 200 mmHg, n = 155), and severe ARDS (P/F ratio ≤ 100 mmHg, n = 65).


**ARDS SeverityMP classes**


The mechanical power, expressed as J/min, was calculated using the following formula:(1)$$Mechanical\;Power=0.098*RR*\left\{{∆V}^{2}*\left[\frac{1}{2}*\;{EL}_{rs}+RR\;*\;\frac{\left(1+I:E\right)}{60*I:E}*\;R_{aw}\right]+∆V\;*\;PEEP\right\}$$
where 0.098 is the conversion factor from liters*cmH_2_O to joule, RR is the respiratory rate, TV is the tidal volume, P_Peak_ is the peak pressure, and ∆P_AW_ is the plateau pressure minus the positive end expiratory pressure.

We measured the distribution of MP in the whole population (see Appendix A), which was divided in three tertiles with similar numbers of patients. The tertiles limits were <16.1 J/min (low tertile), the second tertiles 16.1–21.8 J/min (middle tertile) and the third ≥ 21.9 J/min (high tertile). These thresholds were then applied in the ARDS severity_PF_ classes to define how many patients were included in each MP threshold interval for each class.

### 2.3. Outcome Measurements

All the variables were measured during the first 24 h after entry into the ICU.

Respiratory and gas exchange variables were measured at baseline, i.e., at the ventilatory setting decided by the attending physician substantially in agreement with the protective lung strategy (lower tidal volume). The CT scan (computed tomography scan) variables were collected immediately afterwards, but in standard static conditions, at 5 and 45 cmH_2_O pressure.

The following variables were collected:Gas exchange: Arterial PaO_2_ (partial pressure of arterial oxygen), PaCO_2_ (partial pressure of arterial carbon dioxide), PaO_2_/FiO_2_, and pH were measured, and the ventilatory ratio was calculated using the following equation: [minute ventilation (L/min) × PaCO_2_ (mmHg)]/(predicted body weight × 100 × 37.5 mmHg).Ventilatory setting: tidal volume/ideal body weight, minute ventilation, FiO_2_ (fraction of inspired oxygen), respiratory rate, and positive end expiratory pressure were recorded or calculated.Respiratory system mechanics: peak, plateau, and driving pressures (plateau pressure—PEEP) were measured. Respiratory system elastance was computed as the ratio driving pressure and tidal volume. The mechanical power was normalized to the total lung capacity (TLC).CT scan analysis and derived variables: CT images were processed using custom-designed software (Maluna2020^®^) to calculate various CT-derived variables, including the following: **lung weight; total gas volume; overinflated tissue** (ranging from −900 HU to −1000 HU); **normally inflated tissue** (ranging from −500 HU to −900 HU); **poorly aerated tissue** (ranging from −100 HU to −500 HU); **non-aerated tissue** (ranging from +100 HU to −100 HU); total gas volume was measured as the sum of tissue volume (total lung tissue) and total lung capacity.

### 2.4. Statistical Analysis

Continuous variables are presented as means with standard deviations, while categorical variables are shown as percentages. Linear regression was used to assess the relationship between continuous variables (see Appendix A). To compare groups, we used a one-way ANOVA or the Kruskal–Wallis test for continuous data, depending on suitability, with Bonferroni corrections applied. Mortality rates were analyzed using the Pearson’s chi-square test. *Post hoc* comparisons between pairs of groups were carried out using either Student’s *t*-tests or Wilcoxon tests, as appropriate. A *p*-value of less than 0.05 was considered statistically significant. All analyses were performed using RStudio (RStudio Team 2020; RStudio: Integrated Development for R, RStudio, PBC, Boston, MA, URL: http://www.rstudio.com/ (accessed on 20 October 2024)).

## 3. Results

In Table 1, we report the demographic characteristics of the study population. As shown, patients with mild, moderate, and severe ARDS had similar ages, body weights, and overall illness severity, as assessed by SAPS II (Simplified Acute Physiology Score). The mortality rate significantly increased with increasing ARDS severity_PF_.

In Table 2, we present the main lung CT anatomical and physiological differences observed in ARDS severity_PF_ classes. As shown, the amount of lung edema, as assessed by lung weight, progressively increased with a decreasing PaO_2_/FiO_2_ ratio, while the size of the “baby lung”, assessed by total gas volume, progressively decreased. Non-aerated and well-inflated lung tissue followed similar trends to lung weight and gas volume, respectively, while poorly aerated and poorly inflated lung tissue showed no significant differences.

At similar minute ventilation, PaCO_2_ and the ventilatory ratio increased when the PaO_2_/FiO_2_ ratio decreased. Notably, while the tidal volume pro kg was significantly lower throughout the three ARDS severity_PF_ classes, the respiratory rate significantly increased; consequently, the MP was similar throughout the three classes, given the similar PEEP levels.

In Figure 1, we display the distribution of mechanical power (MP) across the three ARDS severity_PF_ classes, as defined by the Berlin criteria. The distribution of MP was similar, irrespective of P/F ratios. In fact, no significant correlation between MP and P/F ratio was found (*p* = 0.186, adjusted R^2^ < 0.01). Notably, within each class, MP distribution was similarly wide. No correlation was identified between the MP and P/F ratio pairs across the two variables.

In Figure 2, we show the impact of MP on mortality within each of the three ARDS severity_PF_ classes. As shown, an increase in MP was associated with a similar increase in mortality across all classes. Specifically, the increment in the mortality risk per unit increase in MP (measured in Joules/min) was 13%, 18%, and 12% in the mild, moderate, and severe ARDS severity_PF_ classes, respectively. In the overall population, a 1-joule/min increase in MP roughly corresponded to a 1% increase in mortality. This figure illustrates that both ARDS severity_PF_ classes and MP are associated with the mortality rate. A logistic regression analysis showed that these two variables were independently associated with outcomes (P/F *p* < 0.001; MP *p* = 0.016); see online supplement for details. Since MP is a composite variable encompassing factors related to ventilator-induced lung injury (VILI), including respiratory system elastance, tidal volume, respiratory rate, and PEEP, we analyzed MP by breaking it down into its main components within each ARDS severity_PF_ class. The results are presented in Table 3. As shown, differences in MP within the ARDS severity_PF_ classes were not primarily due to tidal volume (a cornerstone of lung-protective strategies) but were driven by differences in respiratory rate and PEEP.

## 4. Discussion

In this study, we found that both ARDS severity_PF_ classes and mechanical power were independently associated with patient outcome. For oxygenation, we maintained the traditional distinction between mild, moderate, and severe ARDS, following the thresholds commonly used in ARDS literature.

For mechanical power, we arbitrarily divided the population into tertiles with similar numbers of patients. The thresholds we identified were low ≤ 16.1 J/min, middle between 16.2 and 21.8 J/min, and high ≥ 21.9 J/min. Interestingly, in a large database study where mechanical power was associated with mortality in 8207 patients, the threshold for increased mortality was 18 J/min [13]. This allows us to discuss both the characteristics of oxygenation impairment and mechanical power, as well as their interaction.

### 4.1. Oxygenation Impairment and ARDS Classes

In this study, we found that the PaO_2_/FiO_2_ (P/F) ratio closely reflects the anatomical impairment across the three ARDS severity_PF_ classes, particularly when comparing mild to moderate and severe ARDS. We observed fewer anatomical differences between moderate and severe ARDS. It is important to note that differences in the P/F ratio for similar anatomical impairment likely depend on the distribution of pulmonary blood flow to non-inflated tissue, which may be influenced by factors such as lung capillary collapse, microthrombosis, and the degree of hypoxic pulmonary vasoconstriction (HPV).

Since our data were collected within 24 h of ARDS diagnosis, i.e., early in the disease course, it is reasonable to speculate that the relationship between P/F and anatomy primarily reflects the natural progression of the disease. At this early stage, patients treated with lung-protective strategies likely did not have sufficient time to develop extensive lung lesions.

### 4.2. Mechanical Power and VILI

Mechanical power (MP) encompasses both anatomical lesions that reflect a prerequisite for VILI development, such as the small size of the “baby lung” and increased elastance, as well as ventilator settings chosen by the physician to maintain acceptable CO_2_ levels (typically below 60 mmHg). To achieve these goals while applying lung-protective strategies with low tidal volumes, the respiratory rate often must be increased. In epidemiological studies of randomized trials, MP was found to be associated with a greater risk of weaning failure [14] and morality [15] (with driving pressure and respiratory rate being the major components of this association) [15]. These data are presented for all ARDS classes without differentiating for severity. A different approach taking severity into consideration is by normalizing MP to static compliance of the respiratory system, which has been shown to be more closely related to outcomes in ARDS patients [16]. Our data suggest that this increase in the respiratory rate, along with a modest rise in PEEP to address oxygenation impairment, is the primary reason for the similar mechanical power values observed across different oxygenation classes. Indeed, the significant decrease in tidal volume from mild to severe oxygenation impairment is offset by significant increases in both PEEP and respiratory rate.

As mechanical power is a summary variable, an increase in one component can be offset by a decrease in another. It is important to emphasize that, at least in experimental studies, the total mechanical power is what drives VILI, independent of the variations in individual components. In summary, mechanical power, more than oxygenation, is primarily driven by the need for CO_2_ clearance.

### 4.3. Interaction Between Oxygenation Classes and Mechanical Power Levels

When we investigated the effects of mechanical power within each ARDS severity class, we observed a clear trend: mortality increased proportionally with the level of mechanical power applied. Notably, the difference in mortality within each ARDS oxygenation class—from the lowest to the highest mechanical power level—was roughly a 1% increase in mortality for every 1 J/min increment in mechanical power. It is important to stress that this increase is not detected by tidal volume alone but is strongly dependent on respiratory rate, a variable rarely considered when discussing the development and progression of VILI.

## 5. Study Limitations

This study has several limitations. First, it is retrospective; however, it reflects the anatomical and physiological conditions at baseline before treatment. Second, it spans a wide timeframe, but we did not account for changes in therapeutic approaches over the past two decades. Our findings are representative of the standard approach, focused primarily on protective lung ventilation, where the goal was to maintain PaCO_2_ below 60 mmHg.

## 6. Conclusions and Future Perspectives

Our findings suggest that respiratory therapy in ARDS should be tailored more to respiratory mechanics and CO_2_ clearance than to oxygenation levels alone. While the available evidence generally indicates that oxygenation impairment primarily reflects the anatomical and natural course of the disease, VILI is mainly driven by the ventilator settings chosen by physicians.

We propose that a possible approach for patients admitted with ARDS should first involve assessing “baby lung” size (using imaging or gas dilution techniques), followed by the application of standard ventilation settings, such as 6 mL/kg tidal volume, 15 breaths per minute, 10 cmH_2_O of PEEP, and prone positioning. If, to achieve the desired PaCO_2_ and PaO_2_ levels, the mechanical power exceeds a threshold—tentatively 18 J/min—alternative forms of respiratory support, such as artificial organs, should be considered.

## Figures and Tables

**Figure 1 jcm-13-07084-f001:**
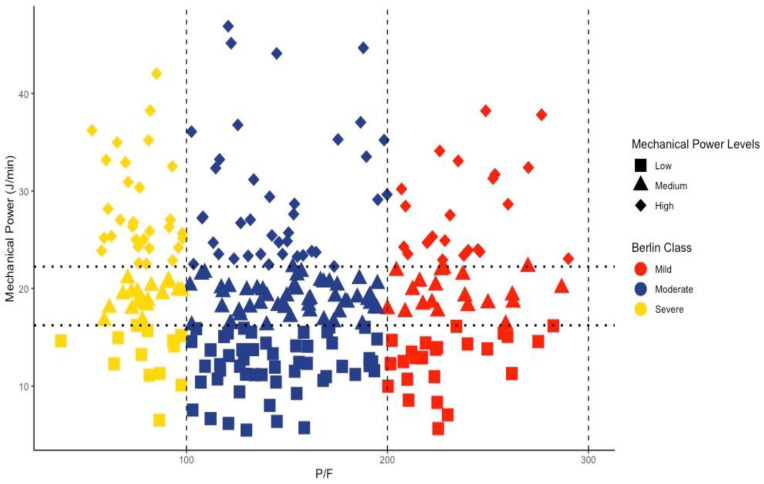
Distribution of mechanical power across the three ARDS severity_PF_ classes, as defined by the Berlin criteria.

**Figure 2 jcm-13-07084-f002:**
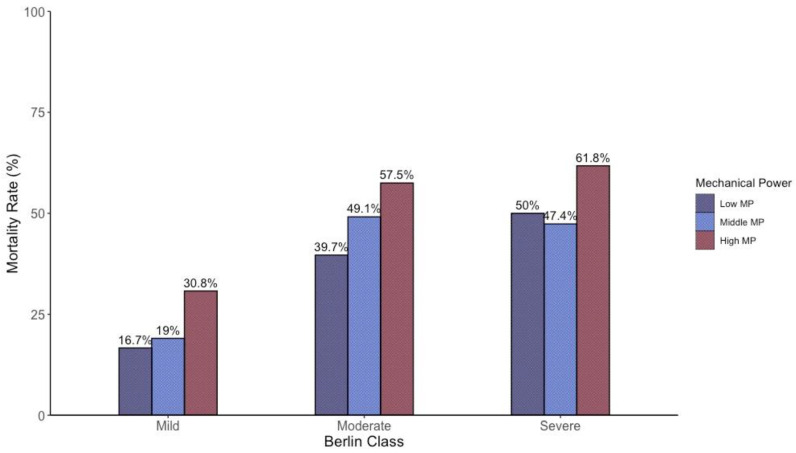
Impact of mechanical power on mortality across the three ARDS severity_PF_ classes.

**Table 1 jcm-13-07084-t001:** Demographic characteristics of the study population and the mortality rate within the study population.

		Berlin Class		
	Mild ^1^	Moderate ^1^	Severe ^1^	*p*-Value ^2,3^
	N = 71	n = 155	n = 65	
**Age, years**	62 (16)	62 (16)	59 (16)	0.5
**Sex**				0.8
Female	23 (32%)	48 (31%)	7 (70%)	
Male	48 (68%)	107 (69%)	42 (65%)	
**Body weight (kg)**	74 (16)	77 (21)	85 (21)	0.054
**BMI**	26 (5)	27 (7)	29 (7)	0.056
**SAPS II**	41 (13)	41 (15)	39 (12)	0.8
Unknown	10	26	6	
**Mortality, n (%)**				<0.001
Dead	18 (23%)	74 (48%)	36 (55%)	

^1^ Mean (SD); ^2^ Kruskal–Wallis rank sum test; ^3^ Pearson’s Chi-squared test for categorical data.

**Table 2 jcm-13-07084-t002:** Lung CT anatomical and physiological differences observed in ARDS severity_PF_ classes.

	Berlin Class			
	Mild ^1^	Moderate ^1^	Severe ^1^	*p* Value ^2^
	n = 71	n = 155	n = 65	
**Lung Anatomy**				
Lung weight, g	1.335 (355)	1.562 (527)	1.927 (829)	<0.001
Total gas volume, mL	1.275 (622)	1.163 (665)	949 (650)	0.004
Overinflated tissue, %	0.20 (0.47)	0.30 (0.87)	0.22 (0.56)	0.93
Well inflated tissue, %	31 (13)	25 (14)	15 (10)	<0.001
Poorly inflated tissue, %	30 (9)	30 (12)	30 (11)	0.64
Not-inflated tissue, %	39 (15)	45 (17)	54 (14)	<0.001
**Physiological Variables**				
PaO_2_/FiO_2_, mmHg	234 (23)	148 (27)	78 (13)	<0.001
PaCO_2_, mmHg	40 (7)	46 (11)	61 (22)	<0.001
Minute ventilation, L/min	8.48 (2.07)	8.11 (2.46)	9.04 (2.28)	0.8
Ventilatory ratio	1.42 (0.44)	1.57 (0.60)	2.24 (0.98)	<0.001
Respiratory system elastance (EL_RS_), cmH_2_O/L	23 (8)	28 (11)	32 (17)	<0.001
Respiratory rate, bpm	16.1 (4.4)	17 (5)	22.2 (5.9)	<0.001
Vt pro kg IBW, mL/kg	7.4 (1.43)	6.62 (1.84)	5.76 (1.58)	<0.001
PEEP, cmH_2_O	10.8 (3)	10.2 (2.6)	11.7 (3.9)	0.135
Mechanical power, J/min	20 (7)	19 (8)	23 (8)	0.186

^1^ Mean (SD); ^2^ continuous variables are expressed as mean (SD). Linear regression was used to assess the relationship between continuous variables (*p*-value refers to linear regression for continuous variables). A *p*-value of <0.05 was considered statistically significant. IBW: Ideal Body Weight

**Table 3 jcm-13-07084-t003:** Respiratory system mechanics, ventilatory settings, and gas exchange differences for different ARDS severity PF classes and ARDS severity MP classes.

	Berlin Class											
	Mild ^1^ (n = 71)				Moderate ^1^ (n = 155)				Severe ^1^ (n = 65)			
	Low Power	Middle Power	High Power	*p* Value ^2^	Low Power	Middle Power	High Power	*p* Value ^2^	Low Power	Middle Power	High Power	*p* Value ^2^
Mechanical power tertiles threshold, J/min	5.49–16.1	16.2–21.8	21.9–46.9		5.49–16.1	16.2–21.8	21.9–46.9		5.49–16.1	16.2–21.8	21.9–46.9	
	n = 24	n = 20	n = 27		n = 58	n = 57	n = 40		n = 12	n = 19	n = 34	
**Respiratory System Mechanics**												
Plateau, cmH_2_O	19.6 (3.8)	24.2 (3.5) *	24.3 (3) *	<0.001	21 (4.3)	23.7 (4) *	26.3 (4.1) * †	<0.001	22 (7)	25 (6)	28 (6) *	0.009
Peak pressure, cmH_2_O	24.1 (4.8)	29.5 (2.8) *	24 (4.4) * †	<0.001	26 (4)	31 (3) *	37 (5) * †	<0.001	26.2 (5.8)	31.8 (3) *	35.5 (6.4) * †	<0.001
Mechanical power, J/min	12 (3)	19 (2) *	27 (5) * †	<0.001	12 (3)	19 (1.55) *	29 (7) * †	<0.001	13 (3)	19 (1.4) *	29 (6) * †	<0.001
Driving pressure, cmH_2_O	10.89 (3)	12.48 (2.60)	12.39 (3.22)	0.126	11.4 (3.4)	13.6 (3.9) *	15.4 (3.1) * †	<0.001	13 (6.8)	12.9 (3.8)	15.2 (4.3)	0.061
Elastance respiratory system, cmH_2_O/L	22 (8)	23 (7)	24 (9)	0.8	27 (13)	28 (10)	31 (9) *	0.030	36 (31)	28 (10)	33 (11)	0.33
**Ventilatory Setting**												
Tidal volume/IBW, mL/kg	7.19 (1.30)	7.73 (1.59)	7.35 (1.42)	0.44	6.34 (1.83)	6.95 (1.60)	6.57 (2.14)	0.108	5.67 (1.95)	5.92 (1.52)	5.70 (1.52)	0.876
Respiratory rate	13.9 (3.5)	15.1 (2.8)	18.9 (4.6) * †	<0.001	14.3 (3.1)	17.1 (4.8) *	20.8 (5.1) * †	<0.001	16.9 (4.4)	16.6 (3.1)	23.4 (5.8) * †	<0.001
FiO_2_	0.42 (0.07)	0.42 (0.06)	0.43 (0.07)	0.5	0.54 (0.11)	0.57 (0.13)	0.61 (0.15)	0.3	0.81 (0.16)	0.78 (0.15)	0.9 (0.1)	0.014
PEEP, cmH_2_O	8.8 (2.4)	11.7 (3.1) *	11.9 (2.6) *	<0.001	9.63 (2.60)	10.17 (2.74)	10.93 (2.5) *	0.039	8.7 (2.6)	11.8 (3.5)	12.6 (4.1) *	0.017
Minute ventilation, L/min	8.48 (2.07)	8.29 (0.84) *	10.15 (1.96) * †	<0.001	6.31 (1.61)	8.30 (1.81) *	10.47 (2.20) * †	<0.001	6.70 (1.36)	7.68 (0.85) *	10.63 (1.86) * †	<0.001
**Gas Exchange**												
pH	7.36 (0.09)	7.35 (0.11)	7.32 (0.11)	0.3	7.4 (0.07)	7.39 (0.07)	7.37 (0.09)	0.2	7.4 (0.06)	7.39 (0.06)	7.4 (0.08)	0.8
PaO_2_, mmHg	95 (15)	100 (16)	103 (16)	0.2	77 (11)	86 (19) *	86 (22) *	0.006	64 (15)	62 (16)	68 (11)	0.4
PaO_2_/FiO_2,_ mmHg	227 (21)	237 (25)	239 (23)	0.14	145 (27)	154 (26)	145 (28)	0.11	80 (17)	79 (11)	77 (13)	0.5
PaCO_2_, mmHg	41 (7)	39 (7)	40 (7)	0.8	47 (9)	45 (11)	47 (13)	0.3	51 (13)	71 (27) *	59 (20)	0.049
Ventilatory ratio	1.20 (0.32)	1.30 (0.22)	1.69 (0.53) * †	<0.001	1.27 (0.41)	1.59 (0.52) *	1.99 (0.70) * †	<0.001	1.50 (0.31)	2.29 (1.0.9) *	2.47 (0.96) *	<0.001

^1^ Mean (SD); ^2^ continuous data are expressed as mean (SD). Continuous data were compared ujsing one-way ANOVA or Kruskal–Wallis test, as appropriate. *Post hoc* analysis was performed by pairwise Student’s *t* test or Wilcoxon test with Bonferroni correction; *: significant at comparison with mild–moderate; mild–severe; †: significant at comparison with moderate–severe.

## Data Availability

The original contributions presented in the study are included in the article/Appendix A, further inquiries can be directed to the corresponding author.

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
