# Peer review of "Does the Intensity of Therapy Correspond to the Severity of Acute Respiratory Distress Syndrome (ARDS)?"

_jcm, 2024, doi:10.3390/jcm13237084_

Round 1

Reviewer 1 Report

Comments and Suggestions for Authors

The paper was well written, and I had some questions:

1.About the initial survey on mechanical power, what about the use of pain control/sedation/muscle relaxant? Was it possible that the setting might influence the survey of mechanical power?

2. As the enrolled periods traced from 2003 to 2024, the possible treatment strategies might differ, such as the use of Dexamthasone for ARDS, the light sedation strategy, antibiotics ....Could they become the possible bias for the mortality?

3. About the Table 1, the demographic data showed some characteristics only. Could the authors add more data as BMI, SOFA scores, APACHE II or co-morbidities?

4.Did the authors survey the different types of ARDS, such as hyperinflammation or hypoinflammation type?

5. On page 6, the authors mentioned about logistic regression analysis showed that P/F and MP were independently associated to outcomes. Could the authors list the other possible variables (a new table or supplement file) that affect the outcome, such as ventilatory ratio, SAPS II, driving pressure...

Author Response

The paper was well written, and I had some questions:

1.About the initial survey on mechanical power, what about the use of pain control/sedation/muscle relaxant? Was it possible that the setting might influence the survey of mechanical power?

R1A1:we thank the reviewer for this question. All patients were sedated and paralysed at the time of study.

  1. As the enrolled periods traced from 2003 to 2024, the possible treatment strategies might differ, such as the use of Dexamthasone for ARDS, the light sedation strategy, antibiotics .... Could they become the possible bias for the mortality?

R1A2: Thank you for the question. Yes, this is possible and indeed we have acknowledge this important point in our discussion where we say:

Second, it spans a wide timeframe, but we did not account for changes in therapeutic approaches over the past two decades

  1. About the Table 1, the demographic data showed some characteristics only. Could the authors add more data as BMI, SOFA scores, APACHE II or co-morbidities?

R1A3: We thank the reviewer for the suggestion, we introduce in the table 1 the BMI. We don’t have the sofa score and the APACHE II; we use SAPS II to assess the severity of the disease in the first 24 h.

4.Did the authors survey the different types of ARDS, such as hyperinflammation or hypoinflammation type?

R1A4: We thank the reviewer for the comment. We did not have the possibility to distinguish between the two subphenotypes as we do not have the markers of inflammation for all patients as theseea are not done routinely in clinical practice. The main article investigating these, published by Calfee, came out in 2014, while our data are from 2003.

  1. On page 6, the authors mentioned about logistic regression analysis showed that P/F and MP were independently associated to outcomes. Could the authors list the other possible variables (a new table or supplement file) that affect the outcome, such as ventilatory ratio, SAPS II, driving pressure...

R1A5: We thank the reviewer for the suggestion. We have included in the supplement the logistic regression analyses between P/F and Driving Pressure, P/F and Ventilatory Ratio, and P/F and SAPS II, with the corresponding equations and p-values.

Reviewer 2 Report

Comments and Suggestions for Authors

Thank you for an opportunity to assess your interesting, retrospective study which further tried to answer the question whether mechanical power (it increase) contributes/is associated with increase mortality in patients suffering from ARDS. Their results are very interesting and novel showing that increased power is associated with increased mortality irrespectively of severity of ARDS.

Overall, manuscript is well written and I have minor suggestion:

1. Please consider changing the title of of your manuscript. In current form it is quite general and does not fully reflect on what study was investigating

2. Please shorten your introduction, some pieces of information could be just referenced since it reflects on generally accepted knowledge

3. Please consider rewriting your discussion. In current form in several paragraphs authors are repeating their results.I would suggest following format

- Main results of your investigation

-How do your findings compare to other similar studies?

-Clinical significance of your results

-limitations

-Conclusions 

Author Response

Thank you for an opportunity to assess your interesting, retrospective study which further tried to answer the question whether mechanical power (it increase) contributes/is associated with increase mortality in patients suffering from ARDS. Their results are very interesting and novel showing that increased power is associated with increased mortality irrespectively of severity of ARDS.

Overall, manuscript is well written and I have minor suggestion:

  1. Please consider changing the title of of your manuscript. In current form it is quite general and does not fully reflect on what study was investigating

R2A1: We appreciate the reviewers’ suggestion; however, we believe the current title is concise and effectively captures the essence of our study’s message.

Please shorten your introduction, some pieces of information could be just referenced since it reflects on generally accepted knowledge

R2A2: We appreciate the reviewers’ comment. However, the introduction is currently 369 words, and we find it challenging to provide a sufficiently informative background in fewer words.

  1. Please consider rewriting your discussion. In current form in several paragraphs authors are repeating their results.I would suggest following format

- Main results of your investigation

-How do your findings compare to other similar studies?

-Clinical significance of your results

-limitations

-Conclusions 

R2A3: We thank the reviewer once again for the suggestion. We do follow the same structure, with subdivisions by theme to enhance the comprehensiveness of each section and maintain alignment with the flow of the results section